# Wettability after Artificial and Natural Weathering of Polyethylene Terephthalate

Annegret Benke [1,*], Johanna Sonnenberg [1], Kathrin Oelschlägel [2], Markus Schneider [1], Milena Lux [3] and Annegret Potthoff [1]

[1] Fraunhofer Institute for Ceramic Technologies and Systems IKTS, Winterbergstrasse 28, 01277 Dresden, Germany
[2] Topas GmbH, Gasanstaltstrasse 47, 01237 Dresden, Germany
[3] School of Engineering Sciences, Faculty of Mechanical Science and Engineering, Technische Universität Dresden, 01062 Dresden, Germany
* Correspondence: annegret.benke@ikts.fraunhofer.de

**Abstract:** The weathering of plastics is always accompanied by a change in surface properties, especially wettability in the case of water. For plastics weathering in an aquatic environment, wettability plays an important role in transport, sedimentation, and dispersion in the water body. To quantify wettability, contact angle measurement is a fast and convenient method that requires little experimental effort. This technique was used with the aim of systematically discussing how measured values of contact angles can contribute to the assessment of the weathering state. Using polyethylene terephthalate (PET), wetting was analyzed on samples from artificial weathering and from controlled, natural weathering. Surface analytical methods were used (Fourier transform infrared spectroscopy (FTIR), scanning electron microscopy (SEM), ultraviolet and visible light spectroscopy (UV/VIS)) to analyze the parameters affecting the contact angle: (i) chemical bond breaking and formation, (ii) eco-corona formation and biofilm growth, and (iii) change in surface structure and roughness. It was found that wettability with water increased during weathering in all cases. The reasons for this varied and depended on the method of weathering. The improved wettability during artificial weathering was due to changes in the polymer surface chemistry. In natural weathering, however, the formation of eco-corona and biofilm was responsible for the changes.

**Keywords:** polyethylene terephthalate; wettability; contact angle; artificial weathering; natural weathering; Fourier transform infrared spectroscopy; yellowness index

## 1. Introduction

The weathering of plastics is an issue that is increasingly being addressed by science. This involves both identifying plastic and microplastic in our environment and understanding the chemical (e.g., photooxidation, hydrolysis), physical (e.g., fragmentation), and biological (e.g., biodegradation) processes that occur under environmental conditions during aging [1–5]. Weathering is always accompanied by changes in plastic surface properties, especially the wettability in the case of water. For plastics that weather in an aquatic environment, wettability plays an important role for transport, sedimentation, and the spread in the water body. While original polymers often show hydrophobic surface behavior, the surfaces become more hydrophilic and therefore more wettable with water during weathering. This can have various causes: the formation of polar functional groups because of photooxidative, hydrolytic, or biodegradation processes, the adsorption of molecules, e.g., leachates, the formation of an eco-corona, and subsequent biological growth.

A suitable technique for quantifying the wettability of a solid surface by a liquid is to measure the contact angle [6–8]. It is a method of little experimental complexity, placing a drop of liquid on a solid surface and measuring the contact angle in balance at the three-phase point between the solid, liquid, and gaseous phases. Small contact angles

(<<90°) correspond to high wettability, and large contact angles (>>90°) correspond to low wettability. In addition to the surface tension of the liquid, the contact angle is influenced by the chemical properties of the solid surface, such as polar and disperse interactions, and its physical properties, such as roughness. Hence, it is a method to obtain a sum of information about the chemical and physical status of the surface and about the wettability in an aqueous environment. In addition, it is possible to calculate the surface free energy and its fractions from contact angle measurements with several liquids, which quantifies the wetting characteristics of the solid surface. For polymers, the model of Owens–Wendt-Rabel–Kaelble (OWRK) is usually used, which distinguishes between polar and disperse components of the surface free energy [9]. The formation of polar groups leads to an increase in the polar fraction of the surface free energy and thus also in the total surface free energy.

In the literature on plastic weathering, contact angle studies are not often performed. They are found in the context of laboratory weathering experiments [7,10,11], in studies of biofilm and eco-corona formation [12,13], and in investigations of polymer biodegradation [14].

In studies of artificial weathering with purified water and ultraviolet (UV) light, the contact angle is especially determined by the chemical properties of the polymer surface and, mostly in later weathering phases, by physical properties such as roughness and heterogeneities. The initial weathering phase is always associated with a decrease in the contact angle due to the production of functional polar groups [7,10,11,15,16]. For polyethylene terephthalate (PET), carboxylic acid end groups (-COOH) are known to be produced during photodegradation. The decrease in the contact angle of low-density polyethylene (LDPE) and polypropylene (PP) during weathering time is due to the formation of carbonyl -C=O groups and -C-O vibrations in ether, carboxyl, and hydroxyl groups (-OH). The authors considered the contact angle as highly sensitive to changes in chemical composition. Julienne et al. were even able to show that measurements of the water contact angle could indicate chemical changes as a sign of weathering on the polymer surface earlier than surface spectroscopic methods.

If the weathering occurs in natural water or in water containing chemical organic or biological components, two processes can affect the wettability of a polymer. On the one hand the contact angle is influenced by adsorbed organic molecules, the conditioning film or eco-corona, or a biofilm. Contact angle measurements of the surface layers show reduced hydrophobicity in comparison to new reference material [12]. On the other hand, colonization by microorganisms can lead to biodegradation. The fact that biodegradation improves wettability, which is caused by chemical changes on the polymer surface, was shown in a study on the biodegradability of PE mulch films by bacteria [14]. Contact angle measurements were also used to demonstrate that the interfacial wettability of the substrate influences the conditioning film and has control over bacterial adsorption and attachment even after conditioning films are established [13,17].

The aim of our study is to systematically discuss how measured values of contact angles can contribute to the assessment of the weathering condition of polymers. Contact angles describe the wettability of the surface. The wettability can be considered as a level for weathering, but we have found that surfaces with the same contact angle value can be in very different weathering states. This is because contact angles are always measured on the outer layer of the surface. Depending on the weathering conditions, this surface can be the polymer itself or a formed eco-corona or biofilm (Figure 1).

The term "weathering", which is interpreted differently by authors, shall have the following meaning in our discussion: There are changes in the physical and chemical properties of a material over time which are triggered by contact with the environment, such as atmosphere, water, and solar radiation. This means that the formation of a new surface by an eco-corona or biofilm is also part of weathering.

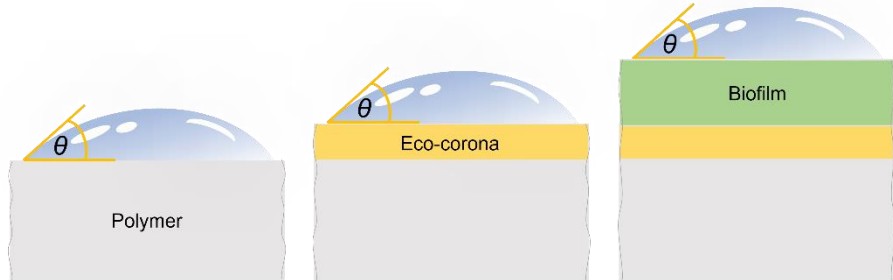

**Figure 1.** Schematic representation of surfaces resulting from the weathering of polymers: polymer surface, weathered by photooxidation, hydrolysis, or biodegradation, eco-corona layer, and biofilm layer. The outer layer is decisive for the measurement of wettability.

To assess the significance of contact angle measurements in weathering experiments, the contact angles were measured comparatively on artificially weathered and on controlled, naturally weathered PET films. The influence of the biofilm on surface wettability was included by measuring the contact angles before and after biofilm removal. Most of the study was then concerned with surface analytical methods for accurately identifying the chemical-physical state of the surface and thus the causes of the corresponding contact angle value or wetting state. Chemical changes were investigated using Fourier transform infrared spectroscopy (FTIR), and the surface was imaged using scanning electron microscopy (SEM). In addition, the yellowing of the samples was determined using ultraviolet and visible light spectroscopy (UV/VIS), another interesting sample property that can be caused by chemical and biological attacks. The results of these methods were considered together to systematically discuss the causes of contact angle changes and to understand the potential of contact angle measurements in assessing weathering.

All tests were performed on PET sheets as an example. PET was selected because of its relevance for commercial applications such as food packaging and polyester fibers. A study of Ziajahromi reported such fibers resulting from the washing process of synthetic clothing, besides PE particles, as being the most frequently detected microplastics across wastewater treatment plants [18]. The production of PET bottles increased worldwide: it almost doubled from 2004 to 2021 to 583.3 billion bottles [19]. Relevant pathways of PET degradation in the natural environment are photooxidative and hydrolytic degradation [20–22].

## 2. Materials and Methods

### 2.1. PET Sheets and Chemicals

PET was purchased from Goodfellow GmbH (Hamburg, Germany), with a size of $10 \times 10$ cm$^2$ and a thickness of 1 mm. According to the manufacturer, the material is largely additive-free. The material is amorphous, with a density of 1.33 g/cm$^3$.

The chemicals used for the preparation of the standard artificial seawater (SAS) were NaCl (Carl Roth GmbH + Co. KG, Karlsruhe, Germany), MgCl$_2$ $\times$ 6 H$_2$O (VWR International GmbH, Darmstadt, Germany), KCl, MgSO$_4$ $\times$ 7 H$_2$O, CaCl$_2$ $\times$ 2 H$_2$O, and NaHCO$_3$ (all from Merck KGaA, Darmstadt, Germany). The total salt content of the SAS was 6.05 g/L, with NaCl 65.6%, MgSO$_4$ $\times$ 7 H$_2$O 16.6%, MgCl$_2$ $\times$ 6 H$_2$O 12.1%, CaCl$_2$ $\times$ 2 H$_2$O 3.8%, KCl 1.4%, and NaHCO$_3$ 0.5%.

### 2.2. Artificial Weathering

For artificial weathering experiments, the PET sheets were cut to a size of $2 \times 5$ cm$^2$, rinsed with ethanol and ultra-pure water, and dried. The resulting samples were each placed in a beaker with 200 mL SAS and covered with a quartz petri dish. During the aging experiment, they were permanently swiveled and irradiated. In addition, the samples were weathered in air with normal humidity and without swiveling. Permanent illumination was achieved by lamps with a spectrum in the ultraviolet and visible ranges and with the special feature of a strong UVB component at approx. 300 nm (Reptile UVB 200, 25 W,

Exo Terra, Hagen, Germany). Irradiance in the spectral range of 280–315 nm was approx. 220 mW/m$^2$ for samples with a 3 cm fluid column of SAS and a distance of 13 cm to the lamp and approx. 700 mW/m$^2$ for weathering in the air with a small distance of 5 cm to the lamp. A waterproof UV sensor (UV-Cosine_UVI, sglux, Berlin, Germany) was used to measure UVB irradiance in the above-mentioned spectral range. The irradiation with UVB light during the weathering periods was calculated as 266 kJ/m$^2$ for 14 days in SAS, 532 kJ/m$^2$ for 28 days in SAS, 1597 kJ/m$^2$ for 84 days in SAS, and 1693 kJ/m$^2$ for 28 days for weathering in air. Beakers with the samples and lighting were housed in a laminar flow box. Constant ventilation kept the temperature within a range of 25 to 28 °C. The complete experimental setup is shown in a conference paper [23]. After 14, 28 and 84 days, the samples were taken from the SAS, rinsed with ultrapure water, dried, and used to analyze aging. The weathering in air was finished after 28 days.

### 2.3. Controlled Natural Weathering

Weathering under controlled natural conditions could be observed during a five-week cruise from 30 May 2019 until 5 July 2019 with the research vessel "Sonne" across the Pacific Ocean from Vancouver to Singapore. The PET sheets were prepared by rinsing with ethanol and ultra-pure water and then fixed on the deck in two mesocosms (one open and one closed) and exposed to the natural conditions of pacific water and solar radiation (open mesocosm only). The samples were constantly flooded with fresh seawater, which was pumped up from the ocean. After various periods of weathering, the samples were taken, individually packed, and kept refrigerated, moist, and dark until analysis. UVB irradiance in the spectral range of 280–315 nm and in the exact sample position under water was measured using same UV sensor as described above. The irradiance fluctuated a lot due to the position of the sun, the partial attenuation by the clouds, and chromophoric dissolved organic substances in the seawater, as well as the rolling of the ship. Average values are approx. 400 mW/m$^2$, and peak values up to 1970 mW/m$^2$ were detected. In contrast to artificial weathering in SAS with permanent irradiation, the samples under controlled natural conditions were exposed to the cycle of day and night. The values for the irradiance are higher but only available at certain times. The irradiation with UVB light was calculated as 484 kJ/m$^2$ for 28 days in the open mesocosm, which is slightly less than the irradiation calculated for 28 days for artificial weathering.

### 2.4. Removing the Biofilm

During natural weathering, a biofilm had grown on the PET surfaces. First, contact angle measurements were carried out on the samples with biofilm on the biofilm layer. Then, the biofilm was removed, and contact angle measurements were carried out again, as well as the other analytical methods (FTIR spectroscopy, UV/VIS spectroscopy). For cleaning, the samples were incubated for 10 min in 3% $H_2O_2$, followed by rinsing with ultrapure water and gently wiping with a soft cloth to avoid damaging the surface as much as possible. Using scanning electron microscopy, it could be shown in several places that the biofilm could be removed. Both the original and weathered samples were resistant to this cleaning procedure. The FTIR spectra before and after cleaning did not differ significantly.

### 2.5. Contact Angle Measurement

Contact angle measurements were carried out with a drop shape analyzer system DSA 100 (Krüss, Hamburg, Germany) according to DIN EN ISO 19403-2. The test liquid was water: SAS was diluted 1:10 with ultrapure water and had a surface tension value of 72 mN/m, which is normal for water. The contact angles were taken in a state of equilibrium and with a minimum number of three replicates for each sample. The samples from the mesocosms and from the weathering in SAS were duplicates; the samples weathered in air were five pieces. However, these were small (2.25 cm$^2$), so only one drop could be placed on each of the highly hydrophilic surfaces.

*2.6. FTIR Spectroscopy*

FTIR spectra were recorded using a Tensor 27 FTIR spectroscope (Bruker Optik GmbH, Ettlingen, Germany) with a diamond ATR crystal (attenuated total reflection). The measurements with a spectral resolution of $4\,\text{cm}^{-1}$ were performed in the range of 400 to $4000\,\text{cm}^{-1}$. The angle of incidence was $45°$. For each spectrum, ten measurements with 32 scans each were averaged.

*2.7. UV/VIS Spectroscopy*

To quantify the yellowing of the samples, their absorption was measured using a Cary 4000 spectrophotometer (Agilent (previously Varian), Santa Clara, CA, USA) with CIE C illumination and a spectral white background. The yellowness index (YI) was calculated according to ASTM 313-20. The absorption measurement and the calculation of the YI were carried out on two samples of the sample set at eight points each to determine the standard deviations of the YI. The maximum standard deviation was 4.6%. As the measurement spot covered a relatively large area ($3\,\text{mm}^2$), the calculated error could be transferred to the other samples.

*2.8. Scanning Electron Microscopy*

To detect changes in surface structure and roughness, polymer surfaces were scanned with field emission scanning electron microscopes (NVISION 40 and ULTRA 55, Carl Zeiss Microscopy GmbH, Jena, Germany). The samples were sputtered with platinum.

## 3. Results

*3.1. Contact Angle Measurement*

The change in the contact angle of PET sheets as a function of the weathering time during the period of 28 days in the two mesocosms on the Pacific Ocean is shown in Figure 2. The distinctly different curves for samples with biofilm (A) and samples cleaned of biofilm (B) are striking. Starting from a contact angle of $77.9°$ for the non-weathered material, all contact angles basically decrease during weathering, which is accompanied by a steady improvement in water wettability. The decrease in the contact angle is significantly greater for samples with biofilm than it is for cleaned samples after a weathering period of 12 days. Nevertheless, complete wettability is still not achieved. For sheets with biofilm, no significant differences in the contact angle between the samples with and without UV irradiation were observed during the period investigated. After cleaning the samples, slight differences in the contact angle values are observed depending on the irradiation.

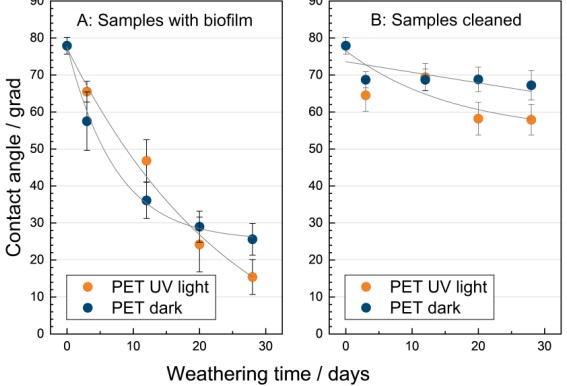

**Figure 2.** Temporal development of the contact angle values of polyethylene terephthalate (PET) sheets weathered under controlled natural conditions in two mesocosms either with (UV light) or without (dark) light irradiation on the Pacific Ocean. Significant decrease in contact angle values for samples taken from the mesocosm with biofilm (**A**). Small decrease in contact angle values for samples with removed biofilm (**B**).

The contact angles for the artificially weathered PET samples also decreased during 84 days of weathering in SAS (Figure 3). For the air-weathered samples, contact angles were measured after 28 days, and complete wettability was observed (contact angle = 0°).

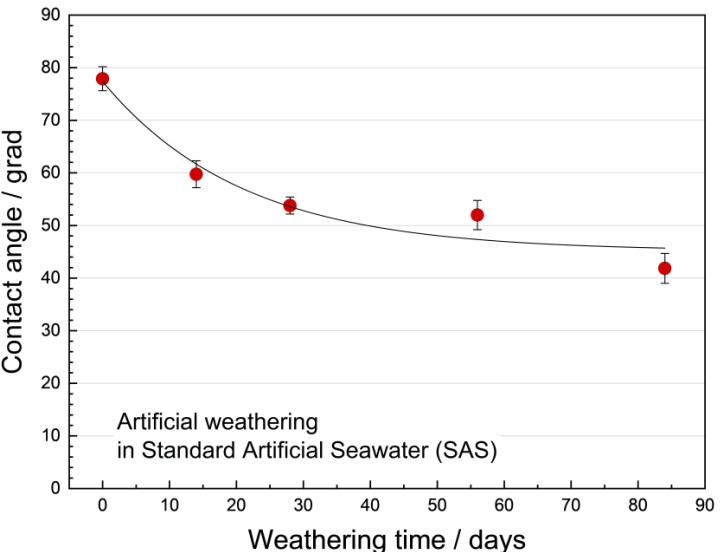

**Figure 3.** Temporal development of the contact angle of PET sheets artificially weathered under laboratory conditions in standard artificial seawater (SAS).

*3.2. FTIR Spectroscopy*

To examine chemical changes on the surfaces, FTIR spectroscopic analyses were performed on both artificially weathered samples and samples weathered in the mesocosms on the Pacific Ocean. In the spectra shown, the laboratory weathered sheets (14, 28, and 84 days), the air-weathered sample, and the naturally weathered samples (28 days, with and without sun light) are compared with the original non-weathered one. The three spectra show essential sections of the FTIR spectrum for PET, the carbonyl region (Figure 4), the fingerprint region (Figure 5), and the hydroxyl region (Figure 6).

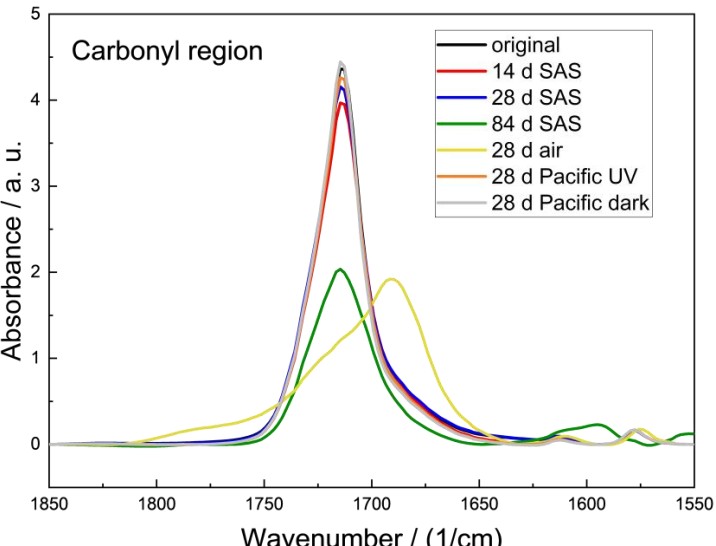

**Figure 4.** Comparison of the carbonyl regions in the Fourier transform infrared (FTIR) spectra of differently weathered PET sheet samples (colored lines) with the original sample (black line). The main maximum decreases due to weathering. The strongest alteration shows the air-weathered sample with a widening of the band and a shift in the main maximum (yellow).

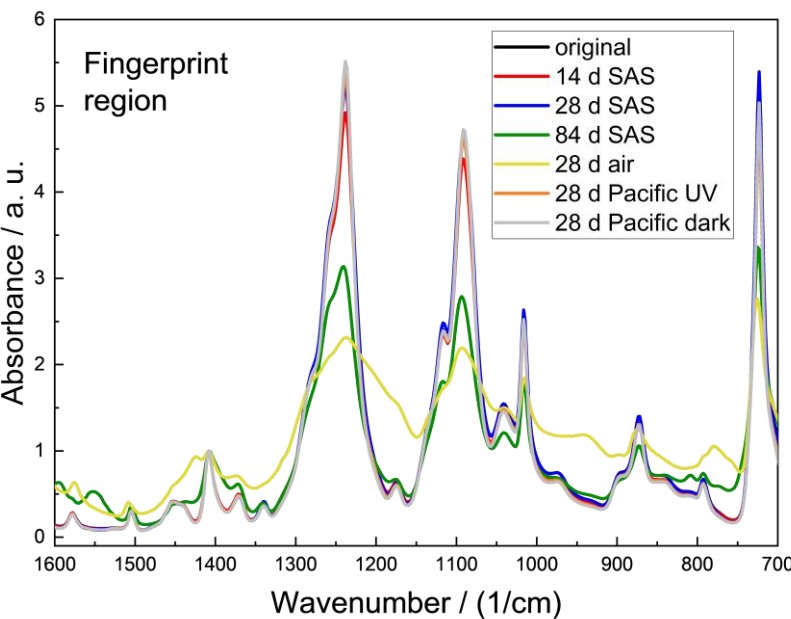

**Figure 5.** Comparison of the fingerprint regions in the FTIR spectra of differently weathered PET sheets (colored lines) with the original sample. The black line of the original sample lies exactly behind the orange one. The strongest signs of weathering are shown by the 84-day weathered sample (green) and the air-weathered sample (yellow) by decreasing absorbance and the widening of the bands.

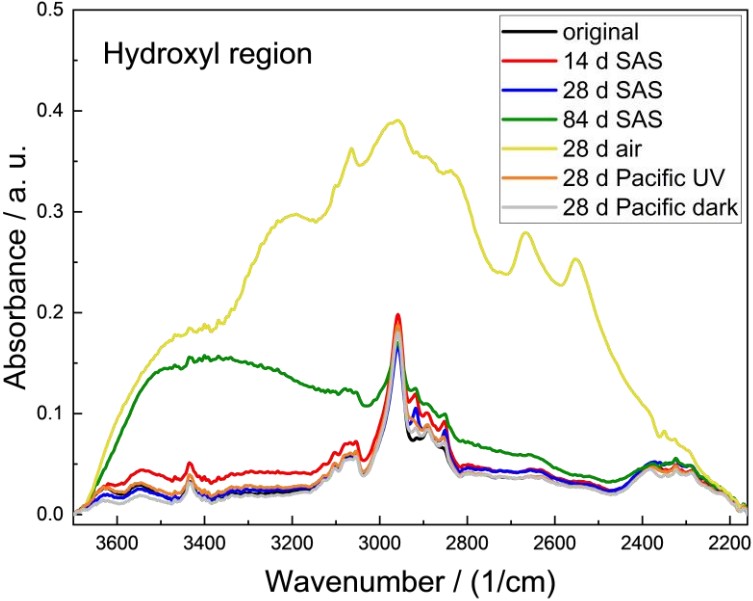

**Figure 6.** Comparison of the hydroxyl regions in the FTIR spectra of differently weathered PET sheets (colored lines) with the original sample (black line). The air-weathered sample differs from the other samples by the increase in absorption in the whole region (yellow).

The analysis of the spectra included correction of the y-shift at 2000 cm$^{-1}$ and normalization with the peak intensity at 1408 cm$^{-1}$ associated with the aromatic skeletal stretching known as the internal IR reference in PET [20,24]. This was used to eliminate measurement-related inconsistencies. For the quantitative evaluation of the peak heights, a linear baseline was subtracted in the hydroxyl region between 2160 cm$^{-1}$ and 3700 cm$^{-1}$, and a non-linear baseline was subtracted in the carbonyl region between 1550 cm$^{-1}$ and 1850 cm$^{-1}$. For samples with a large absorption mountain in the entire hydroxyl range,

where individual peaks cannot be detected due to overlapping, the presence of the peak at 3550 cm$^{-1}$ was checked and verified using peak resolution enhancement (Origin2021).

The carbonyl maximum (C=O carbonyl stretching vibration) decreased with weathering duration, where the maxima of the samples after 14 and 28 days in the laboratory and after 28 days on the Pacific are very close to each other (Figure 4). Chain breaks of the ester bonds resulted in the lower absorption of these bonds. In addition, for the most weathered samples (84 days in SAS and 28 days in air), the first overtone vibration of the carbonyl peak at 3430 cm$^{-1}$ decreased (Figure 6) [10]. The spectra of the samples weathered on the Pacific Ocean were almost not different from that of the original sample. The air-weathered sample showed the strongest weathering phenomena. The broadening of the main peak to larger wavenumbers showed the formation of anhydride groups [25]. The shift of the maximum to 1689 cm$^{-1}$ could represent the formation of quinone groups and the carbonyl stretching vibration of terephthalic acid [10].

Comparing the fingerprint regions of the differently weathered sheets, a similar picture as that in the carbonyl region emerged (Figure 5). The spectra of the original sample, the artificially weathered samples (14 and 28 days), and the samples weathered on the Pacific (28 days) were close to each other. The strong intensities of the C-O stretching vibrations (1240–1270 cm$^{-1}$ and 1080–1150 cm$^{-1}$) were present in all samples, but with slightly decreasing intensities in the 14- and 28-day weathered samples and significantly decreased intensities for the 84-day laboratory weathered samples and the air-weathered sample. For the latter, a broadening of the peak was additionally observed. It was the same with the other strong intensity of the ring vibration at 700–750 cm$^{-1}$. It was present in all samples, but only with much reduced intensity in the two strongly weathered ones (84 days in SAS and 28 days in air). Interestingly, two new peaks at 1550 cm$^{-1}$ and 1593 cm$^{-1}$ in the region of the maxima triplet of ring vibration and C=C stretching vibration (1500–1630 cm$^{-1}$) emerged only in the sample artificially weathered in SAS for 84 days.

The spectrum of the air-weathered sample clearly differed from the others by additional or missing maxima. A new absorption band at 778 cm$^{-1}$ showed the formation of phenyl end groups because of the abstraction of a hydrogen atom from the backbone of the polymer [10]. Another additional peak at 940 cm$^{-1}$ was assigned to the C-H vibration associated with vinyl groups. The peak indicated the photochemical cleavage of carbonyl compounds with the Norrish Type-II Scission route to carboxylic acid end groups [10]. The absence of the peaks at 970 cm$^{-1}$ and 900 cm$^{-1}$ in this sample, in contrast to all other samples, represented the scission of ester links in the polymer chain during photodegradation [10]. Additionally, the peak at 1340 cm$^{-1}$ was missing in this air-weathered sample (CH$_2$ deformation).

In the hydroxyl region, the spectrum of the air-weathered sheet also differed significantly from that of the other samples (Figure 6). The clear increase in absorption in the whole region between 2100 and 3600 cm$^{-1}$ showed the formation of carboxylic acid end groups and hydroxyl components. The main maximum at 2950 cm$^{-1}$ (aliphatic CH$_2$ stretching vibration) and the neighboring peak at 3050 cm$^{-1}$ (aromatic C-H stretching) were present in all samples [10], but with much higher intensity in this one. Furthermore, there were additional maxima indicating carboxylic acid dimers (2550 and 2660 cm$^{-1}$) and carboxylic acid end groups (3290 cm$^{-1}$) [26,27].

The 84-day laboratory weathered sample showed an increased absorption in the range between 3000 and 3600 cm$^{-1}$ due to hydroxylic groups. All other samples—the non-weathered sample, the weathered samples that spent 14 and 28 days in the lab, and the weathered samples that spent 28 days on the Pacific—showed very similar spectra in the hydroxyl region. The two maxima of the alcoholic and aqueous O-H stretching vibration at 3550 and 3630 cm$^{-1}$ were present [28] at the two most weathered samples—the sample that spent 84 days in SAS and the air-weathered sample—with significantly increased intensities.

### 3.3. Yellowness

With the naked eye, a slight to intense yellowing could be seen on all weathered surfaces. To quantify this, the yellowness index was calculated from the instrumentally measured color coordinates (Figure 7). The air-weathered sample showed the most yellowing. Interesting is the fact that the sample weathered in the dark mesocosm also showed slight yellowing.

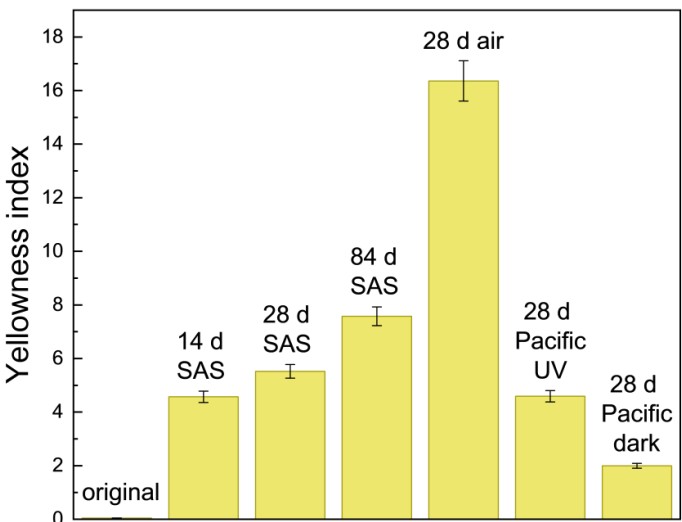

**Figure 7.** Yellowness index illustrating the degree of yellowness of the lab-weathered samples (14, 28, and 84 days in SAS and 28 days in air) and of the 28-day naturally weathered samples with and without UV irradiation after cleaning in comparison with a non-weathered original PET sheet.

### 3.4. Surface Mapping

SEM images showed differences in surface structures for the different artificially and naturally weathered PET surfaces (Figure 8). The original sample was very smooth. During laboratory weathering in SAS, optical roughness increased with weathering time, and a structuring and coarsening of the surface became visible. The sample weathered for 28 days in air showed a surface structure with dimples and elevations.

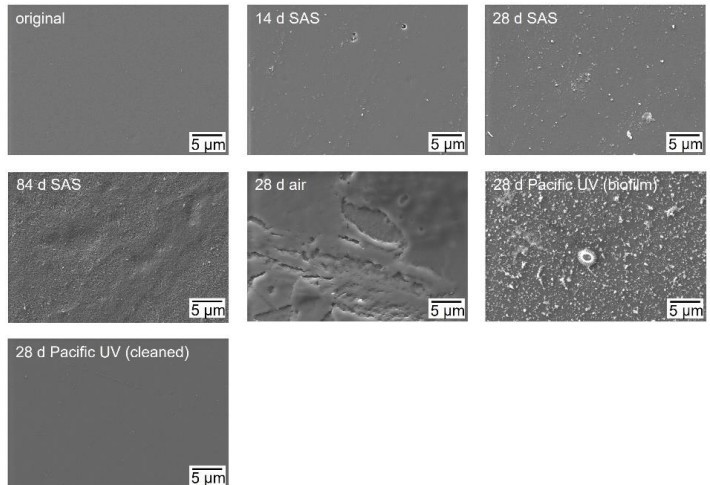

**Figure 8.** Scanning electron microscopy (SEM) images of the lab-weathered samples (14, 28, and 84 days in SAS and 28 days in air) and the 28-day naturally weathered sample (Pacific Ocean under sunlight) in comparison with a non-weathered original PET sheet. The naturally weathered sample is shown as taken from the mesocosm with biofilm on the one hand and after cleaning on the other hand.

After 28 days of weathering in the water of the Pacific Ocean, the entire samples were covered by a biofilm. Diatoms were clearly visible, as well as much smaller structures. The energy dispersive X-ray analysis of these structures showed, in addition to the elements C and O, mainly S and Si, presumably from the biological structures, but also adsorbed Fe, Mg, Al, Zn, Na, Ca, and Cl as components of seawater. After the removal of the biofilm with the above-mentioned method, the surface looked clean, with very few deposits and scratches.

## 4. Discussion

The exponential contact angle decrease after artificial weathering in SAS and in air confirms the results from other scientists; some authors also observed a more linear decrease [7,11,29]. By weathering in artificial seawater and thus eliminating the biological influence, eco-corona and biofilm formation should be excluded. Chemical processes such as photooxidation and hydrolysis, as well as changes in surface structure and roughness, will be examined as reasons for the increase in wettability with water. During weathering in air, the contact angle decreases until complete wettability.

In the literature, the exponential contact angle decrease is shown to be consistent with the increase in surface oxygen content due to the reaction with atmospheric oxygen [29]. As a result, dipole–dipole interactions and hydrogen bonds with the water droplet increasingly develop during the contact angle measurement, and the surface thus becomes more and more hydrophilic.

After controlled natural weathering on the Pacific, the significant decrease in the contact angle from 77.9° to 15.4° for the UV-weathered samples and to 25.6° for the dark weathered samples indicates a strong increase in the hydrophilicity of these samples (Figure 2A). After the removal of the biofilm, the contact angle values also decreased with the duration of weathering, but significantly less so than they did when measured on the biofilm (Figure 2B).

In the following, the possible causes of contact angle changes in PET under our experimental conditions will be discussed: (i) breaking and formation of chemical bonds of the polymer, (ii) eco-corona formation and biofilm growth, (iii) change in surface structure and roughness.

### 4.1. Breaking and Formation of Chemical Bonds of the Polymer

The FTIR spectra of the artificially weathered PET samples are considered with respect to the described indications to understand the increasing hydrophilicity of our samples due to an increasing number of polar functional groups on the surface.

The first point is the striking decrease in the carbonyl peak at 1714 $cm^{-1}$ with the weathering duration in SAS, which shows chain breaks of ester bonds. This is typical for PET aging and is also observed in other studies [10,30–32]. The high reactivity of the carbonyl group is due to the polar double bond between the more electronegative oxygen and the more electropositive carbon. The decrease in the prominent carbonyl maximum is first a contribution to the reduction of the polarity. So, the question arises: how is this consistent with increasing surface polarity? Other reactions must take place that superpose this one. Figure 9 shows the change in absorbance at 1714 $cm^{-1}$ (carbonyl band) and at 3550 $cm^{-1}$ (hydroxyl stretching vibration band) with weathering time. They show opposite behaviors. Both functional groups influence the polarity/hydrophilicity of the sample, which is represented as a sum signal by the contact angle. As compared to the degraded carbonyl groups, the higher polarities of the hydroxyl and carboxyl groups formed during weathering contribute to the overall increasing polarity of the PET surface. Figure 9A shows a decrease in intensities of the carbonyl peak with weathering time for samples weathered in SAS.

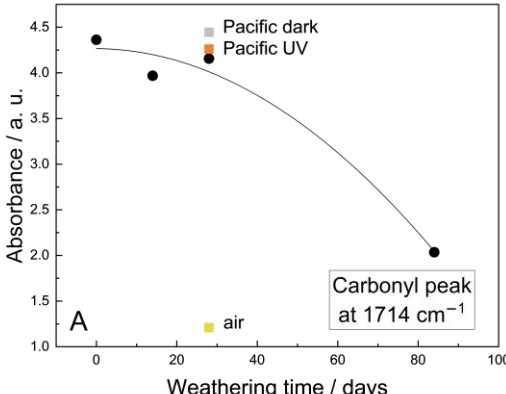 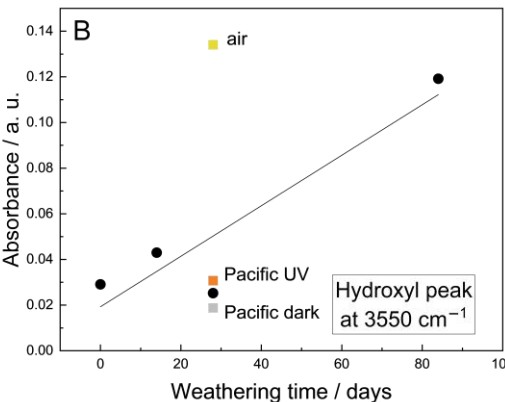

**Figure 9.** Change in functional group intensities (FTIR) over the course of artificial weathering in SAS (black dots with fit curve): decrease in the carbonyl peak at a wave number of 1714 cm$^{-1}$ (**A**) and increase in the hydroxyl peak at 3550 cm$^{-1}$ (**B**). In addition, the 28-day air-weathered sample and the 28-day Pacific-weathered samples are plotted.

The second point is, as just mentioned, the increase in hydrophilicity/polarity through the formation of carboxyl end groups (-COOH), which are polar and very reactive. The partial electron deficiency at the C-atom leads to an even stronger polarization of the polar atomic bond between the O- and the H-atom of the hydroxyl group such that the positively polarized hydrogen atom can easily be split off. In addition, the increase in end groups shows the shortening of polymer chains and thus the degradation of the polymer. In the air-weathered sample, two changes in the carbonyl peak are noticeable: a broadening and a shift in the main maximum. The broadening of the main peak shows the formation of anhydride groups, a combination of two carboxylic acids, and water loss [10,25]. The shift in the maximum to 1689 cm$^{-1}$ can characterize the formation of quinone groups by the integration of oxygen into the aromatic ring such that it is no longer aromatic. It can also show the carbonyl stretching vibration of terephthalic acid, which indicates the decomposition of the polymer into its monomer units [10,17]. The formation of polar double bonds of the carbonyl groups contributes to the increase in polarity. The yellowing of the PET samples, which increases with the duration of weathering (Figure 7), can be explained by the formation of colored quinone groups and by the frequent occurrence of double bonds in the molecule, e.g., carboxylic acid anhydride groups.

The third point is the increase in polarity and hydrophilicity due to the formation of hydroxyl groups, which have a higher polarity than carbonyl groups. In all samples, but especially clearly in the most weathered samples in SAS after 84 days and in air, the intensities of the two maxima of the alcoholic and aqueous O-H stretching vibration at 3550 and 3630 cm$^{-1}$ increase. Figure 9B shows an example of an absorption increase in the hydroxyl region and an increase in intensities with time for the band O-H stretching vibration at 3550 cm$^{-1}$.

The fourth point is a further contribution to the increase in polarity and hydrophilicity caused by the degradation of aromatic, non-polar structures on the polymer surface. The strong intensity of the ring vibration at 723 cm$^{-1}$ decreases for the samples weathered for 84 days in SAS and in air to almost half the value of the non-weathered sample (Figure 5). This decrease in aromatic carbon content is also described by Hurley [29].

The FTIR spectra of samples from controlled natural weathering on the Pacific showed increased absorptions in the range of 3000 to 3800 cm$^{-1}$ (OH and NH vibrations, respectively) and in the fingerprint region, which was attributed to the biofilm. After cleaning the surface, these absorptions were significantly reduced. Only spectra from purified samples were evaluated. The spectra of samples weathered with and without the influence of sunlight showed only minor chemical changes in the polymer (Figures 4–6). The quantitative evaluation of two important bands for PET, the carbonyl band at 1714 cm$^{-1}$ and the hydroxyl band at 3550 cm$^{-1}$, showed absorption values after 28 days of natural weathering

differing only slightly from the values of the non-weathered polymer (Figure 9). That is interesting, because the UVB irradiation in the open mesocosm is about the same as it is after 4 weeks in the laboratory. The cause is the formation of an eco-corona and, based on this, the biofilm under marine conditions. When polymer surfaces are exposed to natural water, containing a complex mixture of organic macromolecules such as humic and fulvic acids, excreted waste products and lipids, polysaccharides, and proteins, an eco-corona begins to form after only a few seconds, growing in layers and varying in thickness from initial flat monolayers to multilayers [33,34]. The surface layer of the eco-corona and biofilm becomes increasingly dense and opaque over time and can almost completely prevent UVA and UVB transmittance and thus photodegradation [35].

It can be summarized that, in the case of artificial weathering, the breaking and formation of chemical bonds make a major contribution to the decrease in the contact angle and the increase in hydrophilicity. In contrast, they have no significance in the case of the natural weathering of the samples.

### 4.2. Eco-Corona Formation and Biofilm Growth

After weathering on the Pacific, increasing biofilm formation with weathering time was visible, with no difference between the samples from light and dark weathering in light microscopy. However, an analysis of the organisms was not carried out. The properties of the surface are determined by the growth of the biofilm, a layer of organic and inorganic material. The increasing hydrophilicity of the sample surface correlates with the biofilm growth during the weathering time and can therefore be explained by this. This correlation is described in the literature, e.g., a 3-week study with plastic bags in the sea at a depth of 2 m [36], and a study on the biofouling of different polymers in marine water [32]. Furthermore, Krause et al. reported no indication of chemical changes in plastic samples after long-term weathering in the sea. Only the contact angles of the surfaces changed significantly compared to the original material [12].

After removing the biofilms of the mesocosm samples, a yellowing of the polymer became visible (Figure 7). This is interesting because there were no spectroscopically detectable chemical changes on the surfaces. We found clear differences in the yellowness index: higher values were measured for the sample weathered under sunlight than for the sample weathered in the dark. In addition to the contact angle, the yellowness index is a sensitive indicator for surface weathering. We found a very good correlation of the yellowness index with the contact angles for all PET samples examined here (Figure 10). This is an interesting result, as yellowing and wettability are not directly related.

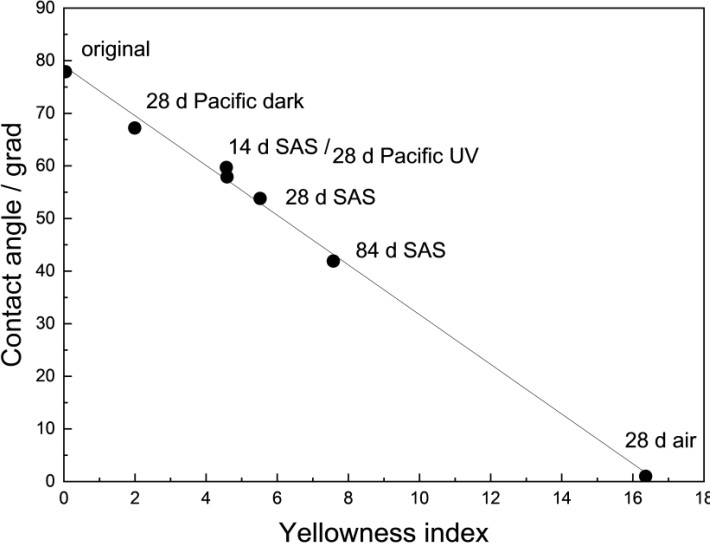

**Figure 10.** Correlation of contact angle and yellowness index for the original sample and differently weathered PET sheets.

Three different reasons for the yellowing of these samples can be discussed: The first reason is an attack by microorganisms, the initial phase of biodegradation. If a biofilm is present, biodegradation due to the action of microorganisms such as bacteria is possible, whereby functional groups and double bonds can be removed and incorporated [37]. PET was known for a high resistance to biodegradation because of its compact structure [38], but in 2016, the microbial degradation pathway in the environment was published [39]. The light-dependence of the yellowing may be due to the UV-induced pre-damage of the polymer and the synergistic relationship of abiotic and biotic weathering [40], as well as to possible different microbial communities. The second reason for yellowing can be seen in a hydrolytic degradation process, which does not require UV light and leads to yellowing [38]. As a third reason, a contribution of the eco-corona and biofilm layer to the yellowing can be suspected, because the layer directly on the polymer presumably cannot be completely removed with our sample cleaning method.

The conclusion is that weathering in a natural environment strongly influences hydrophilicity by developing an eco-corona and biofilm and all related processes on the surface. Changes in the contact angle can therefore be attributed to this.

### 4.3. Change in Surface Structure and Roughness

Is the increasing hydrophilicity additionally influenced by surface structure and roughness? In the scanning electron micrographs of the PET surfaces of the artificially weathered samples, an increasing structuring with weathering time is visible, and the roughness seems to increase (Figure 8). On the one hand, the literature often reports increasing roughness with increasing weathering duration [41–43]. On the other hand, the contact angle studies by Busscher et al. on twelve different polymer surfaces after various surface roughening procedures are known. It was found that changes in roughness had no effect on the contact angle when the smooth surface showed contact angles between 60° and 86° [44]. The contact angle of the original PET surface examined here is 77.9° (Figures 2 and 3) and should therefore not be influenced by roughness. Attempts to measure the roughness of the original PET samples and those weathered in the laboratory with a 3D laser scanning microscope (VK-X200, Keyence, Neu-Isenburg, Germany) resulted in average roughness values (Ra) of Ra < 0.1 μm for all samples. If the Ra of polymer surfaces is very small (Ra < 0.1 μm), it no longer has any influence on the contact angle [44]. It is therefore concluded that an influence of the roughness on the wettability of the lab-weathered PET samples cannot be excluded, but this should be very small.

For the mesocosm samples, the roughness is increased due to the biofilm. An increased microscopic roughness was observed through dried objects, such as skeletons of diatoms (Figure 8). In addition, an irregular macroscopic roughness was visible in the camera image of the contact angle meter due to larger accumulations of biofilm. This affected the contact angle: water droplets were pinned at peaks or dispersed in depressions. These are sources of error that lead to larger standard deviations of the contact angle values than with cleaned samples. Because measuring the contact angle on the dried biofilm layer does not reflect the reality of the wettability of a biofilm, the influence of roughness on the contact angle will not be considered further here. After the removal of the biofilms, the roughness of the samples is very low (Figure 8) and does not affect the contact angle (see discussion above).

In summary, although the contact angle is theoretically affected by the roughness of a surface, the slight change in roughness due to weathering is negligible in our polymer samples. The biofilm leads to a strong decrease in the contact angle. The hydrophilicity of the biofilm has a much greater influence than the roughness.

## 5. Conclusions

Contact angle measurements showed an increasing water wettability of the PET samples after both artificial and natural weathering. We found that the reasons for this were different and depended on the type of weathering. The improved wettability with artificial weathering was due to changes in the chemistry of the polymer surface. With

natural weathering, on the other hand, the formation of eco-corona and biofilm was responsible for the changes.

To identify the actual weathering situation, further analytical methods were necessary in addition to the contact angle measurements. From the changed wettability alone, it was not possible to determine whether the surface changes were caused by chemical, physical, or biological processes. The wettability provides information about the changed hydrophilicity of the surface during weathering but not about the weathering state of the sample. Contact angle measurement is a well-suited method for describing changes in surface wettability very sensitively.

**Author Contributions:** Conceptualization, A.B., K.O. and A.P.; Methodology, K.O., M.S. and A.P.; Validation, A.B., J.S. and A.P.; Investigation, A.B., M.L., K.O. and M.S.; Writing—Original Draft Preparation, A.B.; Writing—Review & Editing, A.B., J.S., K.O. and A.P.; Visualization, A.B. and J.S.; Supervision, A.P.; Project Administration, A.P.; Funding Acquisition, A.P. All authors have read and agreed to the published version of the manuscript.

**Funding:** This research was funded by the German Federal Ministry of Education and Research (BMBF) with the grant number 03G0268TB. The APC was funded by Fraunhofer Gesellschaft e. V.

**Data Availability Statement:** Not applicable.

**Acknowledgments:** We would like to thank Andreas Böhme for the FTIR spectroscopic measurements, Stefanie Hildebrandt for the UV/VIS spectroscopic measurements, Sören Höhn and Sabine Fischer for their support with the scanning electron microscope, and Jana Brinkmann for the realization of the graphical abstract.

**Conflicts of Interest:** The authors declare no conflict of interest.

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
