# Peer review of "Wettability after Artificial and Natural Weathering of Polyethylene Terephthalate"

_environments, doi:10.3390/environments9110134_

Round 1

Reviewer 1 Report

This is an interesting paper and I must confess it is not my natural territory, but I am familiar with degradation and the materials science behind the paper.

As the authors suggest, contact angle is a simple measurement and indeed it is a first year undergraduate experiment.  The equipment utilised in the experiments is clearly a high specification, but having said that, the authors have not said much about the experimental approach in the context of the aim of the paper, especially in the context of the pacific ocean, whether the testing of flat test specimens is appropriate.  The reality is that plastics in the oceans vary from large irregular specimens to microparticles and it is the later that is more important.  As the authors suggest parameters such as roughness matter and so I wonder how this work would relate to measuring contact angle from irregular and small fragments and particles taken from the ocean.  Experimental techniques is important and the authors should recognise this in the paper.  As it turns out, it may not matter as I don't think the paper has the right title that I will explain later.

As mentioned measuring contact angle is not hard (in theory) and so I wondered what really is novel in this work.  Is it really the measurement of the contact angle or is it something else?  Again I will come back to this point.

In setting the scene, the authors should describe the waste polymer inventory in the ocean and set the choice of PET in this context.  I also think they should have chosen at least a second or third polymer to compare results (at least in the lab accelerated weathering tests).  Otherwise, can we be convinced by the utility of the approach?  So I do have an issue with the experimental design and need to understand why real samples from the ocean were not taken and measured as well as a broad selection of polymers.

There is a difference between degradation of the polymer and the creation of the biofilm and this seems to have got confused in the paper.  To what extent are the results a function of the biofilm or the polymer?  Figure 1 suggests the results between PET UV light and dark samples are not dissimilar but is strongly dependent on whether a biofilm is present.  The figure is not entirely clear on error bars and how many samples were tested.  This needs to be improved so the reader can establish whether the two sample populations are truly significantly different.  In that context, I am not convinced that PET UV and dark samples are separate populations and perhaps a meta-analysis may be useful.  Looking at Figure 2, assuming no biofilm is present, the results seem comparable with the cleaned samples which support the points being made.  The air weather results initially appear odd and the FTIR also suggest initially something odd.  This is because samples earlier than 28 days were not tested.  I think it is really important that the additional samples are tested at say 1, 2, 4, 8, 16 before zero contact angle observed at 28 days.  This will help interpret the FTIR in the context and scenario of this paper.

The paper deals more with the FTIR analysis than contact angle and I suspect this is the real novelty and understanding the science behind the degradation.  The authors suggest 'the contact angle was found to decrease with weathering time for all samples...' - this is not precise or really true as it is very dependent on the conditions and whether a biofilm is present.  Figure 9 is the one that hit homes and really suggests the yellowness index is more important than contact angle and that UV/VIS spectroscopy is the way to go which would be more shape independent for samples.  This should be linked to the underlying degradation to FWHM in the FTIR analysis.  In that context, whilst there is more experimental work required to make this paper complete, the title should reflect the content more such as 'characterising the degradation of weathered PET polymer through FTIR and UV/VIS spectroscopy' given the contact angle is really a proxy.  So in summary the narrative may need rethinking as to what this paper really contributes and for those reasons I would suggest major revision with further experimental work.

Reviewer 2 Report

Review report

Can contact angle measurements contribute to assessing the weathering of plastics?

It’s a nice try to say something which is already known phenomenon. Weathering is nothing but degradation or erosion either induced or occurring naturally in a polymer surface. So, contact angle has been used by various articles earlier (follow journals) commonly to show degradation or changes in wettability due to changes in chemical structure of the polymer surface occurred via weathering.

I feel the article is not worth publishing.

Comment 1:

There is very less discussion about the various properties associated with wettability and more about other analysis like FTIR etc.

Comment 2:

Roughness has not been measured, just by visualization we cannot say roughness is responsible for increasing hydrophilicity. In contact angle measurement there are various models which depicts roughness factor which is a term used in mathematical models associated with wettability. These models clearly describe that sometimes increase in roughness increases the hydrophobicity i.e., increase in contact angle and its due to factors like “surface free energy”, which was not measured at all.  

Reviewer 3 Report

Dear Authors

The manuscript is well organized. The problem statements agree with the title and have significance. The methods used to gather the data for this article were clearly explained.  The quality of citations is good , autors have referenced the interesting works in this field of research. The topic is interested and the result are concreted and useful for the scientific community. It is known that the wettability of polymer surfaces changes during the weathering process. In this situation wettability contact angle measurement is a fast and convenient method that re-quires little experimental effort. Thus, the topic is interested, however, some aspects must be revised before the acceptation for this Journal.
Some specific comments are listed below.
1. Abstract: The aim and conclusions of the study must be clearly visible. Please, you emphasize the importance of performed experiments.
3. Charts  need correction (especially rys. 6.  - please let me know from how many trials the mean result was calculated?
3. Please check literature citations  - in accordance with the instructions for authors.

Round 2

Reviewer 1 Report

Reading the response from the authors, I think they have been a bit too defensive to the interpretation of their results (I note there were no changes at all here), but rather the focus on improving why they did things in the introduction and the experimental.  Clearly, this does improve the paper, but I do feel the changes are rather superficial and they have missed an opportunity.  My concerns largely remain.  Some differences are down to perspective and opinion, but as the other reviewer suggests there are some flaws that are perhaps not being tackled.  The title is an improvement, but I feel it still isn't truly reflective of the analysis and conclusions being reached and the main body of the work.  If I am generous, I might say let the paper stand on its merits and if I were harsh I might say insufficient progress and development has been made to create the level of novelty truly needed to publish.  It was interesting that in defence of some points, they made reference to earlier published work which demonstrates this point.  I have recommended minor corrections just to allow the authors to rethink through the earlier points and make changes to the analysis if they deem appropriate.

Reviewer 2 Report

After addressing the reviewer's comment, I feel the article is ready to be published.

Author Response

Dear Reviewer,
Thank you for your renewed assessment. 
We have reviewed the cited references and consider them to be relevant to this paper.